# Whole genome sequencing refines stratification and therapy of patients with clear cell renal cell carcinoma

Richard Culliford [1,27], Samuel E. D. Lawrence[1,27], Charlie Mills [1,27], Zayd Tippu[2,3,4,27], Daniel Chubb[1], Alex J. Cornish [1], Lisa Browning[5], Ben Kinnersley [1,6], Robert Bentham[6], Amit Sud [1], Husayn Pallikonda [2,3,4], The Renal Cancer Genomics England Consortium*, Anna Frangou [7,8,9], Andreas J. Gruber [10], Kevin Litchfield [11], David Wedge [12,13], James Larkin[2,3], Samra Turajlic [2,3,4] & Richard S. Houlston [1] ✉

Clear cell renal cell carcinoma (ccRCC) is the most common form of kidney cancer, but a comprehensive description of its genomic landscape is lacking. We report the whole genome sequencing of 778 ccRCC patients enrolled in the 100,000 Genomes Project, providing for a detailed description of the somatic mutational landscape of ccRCC. We identify candidate driver genes, which as well as emphasising the major role of epigenetic regulation in ccRCC highlight additional biological pathways extending opportunities for therapeutic interventions. Genomic characterisation identified patients with divergent clinical outcome; higher number of structural copy number alterations associated with poorer prognosis, whereas VHL mutations were independently associated with a better prognosis. The observations that higher T-cell infiltration is associated with better overall survival and that genetically predicted immune evasion is not common supports the rationale for immunotherapy. These findings should inform personalised surveillance and treatment strategies for ccRCC patients.

Renal cell carcinoma (RCC) is an increasing global health problem with 431,000 new diagnoses each year, set to increase to 666,000 by 2040[1]. Around 75% of RCCs are clear cell RCC (ccRCC) tumours. These cancers have a variable clinical course and while 75–80% of patients present with apparently localised disease and are offered curative intent treatment 30% will subsequently relapse[2]. There is therefore a pressing need for more accurate risk stratification, to guide clinical decisions relative to therapy and surveillance.

While therapeutic advances in the treatment of metastatic ccRCC have been made with the advent of antiangiogenic targeted therapies and immune checkpoint inhibitors (ICIs) only a subset of patients experience durable clinical benefit. Importantly, clinical biomarkers

[1]Division of Genetics and Epidemiology, The Institute of Cancer Research, London, UK. [2]Renal and Skin Units, The Royal Marsden NHS Foundation Trust, London, UK. [3]Melanoma and Kidney Cancer Team, The Institute of Cancer Research, London, UK. [4]Cancer Dynamics Laboratory, The Francis Crick Institute, London, UK. [5]Department of Cellular Pathology, Oxford University Hospitals NHS Foundation Trust, Oxford, UK. [6]Department of Oncology, University College London Cancer Institute, London, UK. [7]Nuffield Department of Medicine, Big Data Institute, University of Oxford, Oxford, UK. [8]Algebraic Systems Biology, Max Planck Institute of Molecular Cell Biology and Genetics, Dresden, Germany. [9]Algebraic Systems Biology, Centre for Systems Biology Dresden, Dresden, Germany. [10]Department of Biology, University of Konstanz, Konstanz, Germany. [11]Cancer Research UK Lung Cancer Centre of Excellence, University College London Cancer Institute, London, UK. [12]Manchester Cancer Research Centre, University of Manchester, Manchester, UK. [13]NIHR Manchester Biomedical Research Centre, Manchester, UK. [27]These authors contributed equally: Richard Culliford, Samuel E. D. Lawrence, Charlie Mills, Zayd Tippu. *A list of authors and their affiliations appears at the end of the paper. ✉e-mail: richard.houlston@icr.ac.uk

fail to reconcile the variable disease course following surgery[3–5] in patients receiving adjuvant PD1-therapy.

The need to understand ccRCC biology, to inform development of novel therapies and better predict patient outcomes, has been a major motivation in sequencing studies. While these projects have identified recurrent gene mutations and chromosomal rearrangements, analyses have primarily been based on whole-exome sequencing or panel testing of cancer-associated genes, hence the full complement of drivers is incomplete. Correspondingly, studies of the relationship between clinical parameters and genomic alterations have been limited[6–9].

To advance our understanding of ccRCC, we analyse whole genome sequencing (WGS) data from 778 ccRCC patients recruited to the UK Genomics England (Gel) 100,000 Genomes Project (100kGP)[10,11].

## Results

### The Gel cohort

The analysed cohort (100kGP, release v14) comprised tumour-normal (T/N) sample pairs from 778 patients (mean age 63 years, range 25–88 years) with primary ccRCC recruited to 100kGP through 13 Genomic Medicine Centres across England (Fig. 1). Comprehensive clinico-pathology information on the patients is provided in Supplementary Data 1. We restricted our WGS analysis to samples with high-quality data from polymerase-chain-reaction (PCR) free (allowing for accurate uniform coverage without sequencing bias from PCR duplicates), flash-frozen fresh tumour samples ('Methods'). For 29 of the patients, WGS data on multi-regional sampling of tumours was available (2–4 samples per tumour, 94 samples in total). In addition to using variant calls from the 100kGP analysis pipeline we: (i) removed alignment bias introduced by ISAAC soft clipping of semi-aligned reads[12]; (ii) called tumour copy number using Battenberg[13]; (iii) called structural variants (SVs) from a consensus of Manta[14], LUMPY[15], and DELLY[16]; (iv) removed insertion-deletions (indels) within 10 base pairs (bp) of a common germline indel. Complete details on sample curation, somatic variant calling, and annotation of mutations are provided in the "Supplementary Methods".

Restricting our analysis to WGS data on one sample per patient ('Methods'), we identified 4,267,943 single nucleotide variants (SNVs), 699,100 indels, and 19,756 chromosomal rearrangements or structural

variants (Supplementary Data 2). While the median tumour mutational burden (TMB) was 2.07/Mb, three tumours displayed a hypermutated phenotype: i.e., excessively high SNV/indel mutation burden (maximal SNV/Mb = 33.65, maximal indel/Mb = 21.77). Twenty-two of the patients (2.8%) were carriers of pathogenic germline variants in one of the well-established RCC susceptibility genes (*CHEK2, FH, MITF, SDHA, VHL*)[17] and 10 (1.2%) in a pan-cancer susceptibility gene (*ATM, BRCA2, BRIP1, FANCM, MSH6, PALB2, PMS2*; Supplementary Data 3). Four of the 32 patients had a prior history of non-RCC cancer (thyroid, prostate, testicular and chronic lymphocytic leukaemia).

### Driver mutations

Protein-coding driver gene identification at the base pair level was performed using IntOGen[18], which incorporates seven complementary algorithms. A total of 38 genes were identified as driver genes, including 25 well-recognized drivers and 13 which either had not been reported previously or have frequencies <1% in landmark genomics studies of ccRCC[19–23] (Fig. 2). Of the mutations annotated by AlphaMissense[24], 72.4% of driver gene missense SNVs (436/602) were predicted to be pathogenic as compared to 29.8% (11,112/37,297) of the missense SNVs in non-driver genes ($P = 4.2 \times 10^{-112}$). The major known ccRCC driver genes were mutated at close to reported frequencies[19–21,23,25–27]: *VHL* (80.2%), *PBRM1* (49.9%), *SETD2* (17.7%) and *BAP1* (11.8%). Subclonal drivers, such as *TSC1* were, however seen with lower frequency, which is likely to be the consequence of our reliance on a single biopsy rather than multi-regional sampling as per studies such as TRACERx[8] (Supplementary Data 4). All of the candidate coding drivers were detected at low frequencies, as expected given the scale of previous exome sequencing studies (0.3–2.4%; Supplementary Fig. 1a–m, Supplementary Data 4). The increased power of our study has enabled us to assign driver status to a number of genes recurrently mutated at low frequency in other studies (Supplementary Fig. 2).

Mutations in the candidate drivers were frequently accompanied by loss of heterozygosity (LOH) (see 'Recurrent structural and copy number alterations'), implying loss of function (Supplementary Fig. 3). Furthermore, leveraging TCGA[28] and GTEx[29] expression data as well as DepMap gene perturbation screening[30,31] in addition to referencing literature provide supporting evidence that 6/13 of the candidate genes we identify have relevance to the biology of RCC

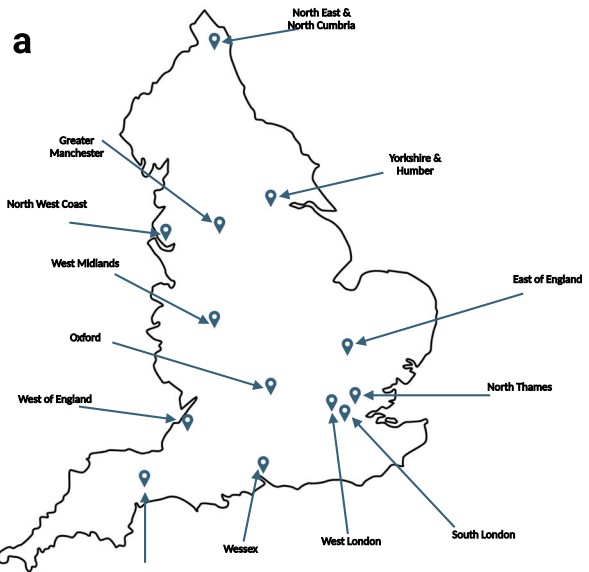

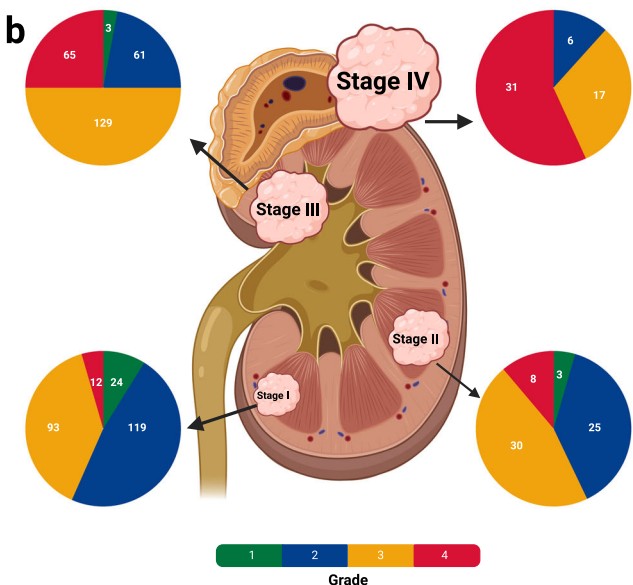

**Fig. 1 | Overview of the Gel cohort of ccRCC patients. a** The location of the 13 Genomic Medicine Centers (GMCs) across England from which patients were recruited. **b** The breakdown of the cohort by tumour grade and stage. Figure 1

created with BioRender.com released under a Creative Commons Attribution-NonCommercial-NoDerivs 4.0 International license.

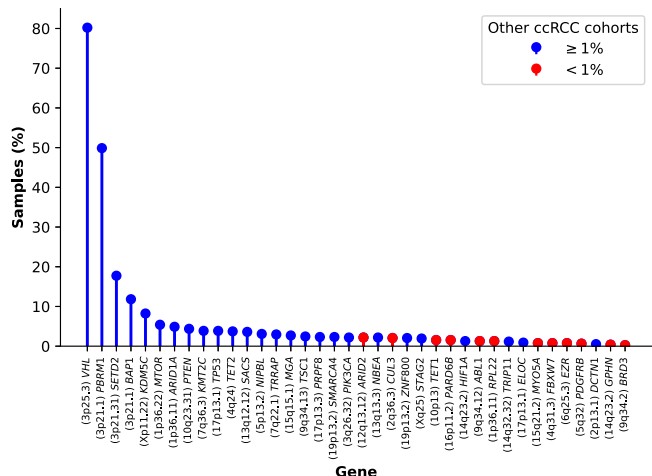

**Fig. 2 | Frequency of patients with nonsynonymous mutations in driver genes.** The cohort frequency of a driver gene reported as being above (blue) or below (red) 1% in other ccRCC cohorts is indicated.

(Supplementary Fig. 4a, b, Supplementary Data 4). Candidate drivers emphasise the central importance of epigenetic modification in the development of ccRCC through SWI/SNF mediated chromatin remodelling (*BRD3, ARID2*), histone deubiquitination (*CUL3, FBXW7*) as well as the role of methylcytosine dioxygenase activity (*TET1*). Biological mechanisms highlighted by the candidate drivers are shown in Fig. 3a–e. These included new vessel formation (*PDGFR-β*), ribosomal activity (*RPL22*), cytoskeletal interactions (*EZR, GPHN, NBEA*), cell division and cell polarisation (*PARD6B*). Mutations of *ARID2, CUL3, TET1, FBXW7, EZR* and *GPHN* were frequently accompanied by copy number loss consistent with their predicted/documented tumour suppressor roles (Supplementary Data 4).

As previously reported[32], mutations in *VHL* and *ELOC*, which both play a key role in oxygen sensing and degradation of hypoxia-inducible factors, were mutually exclusive ($Q = 0.002$). Co-occurrence analysis also supported mutually exclusive relationships: *BAP1* and *PBRM1* ($Q = 4.36 \times 10^{-12}$), SWI/SWF mediated chromatin modifiers *PBRM1* and *ARID2* ($Q = 0.04$), and *SETD2* and *BAP1* ($Q = 0.049$). In contrast, mutations of *ARID2* and *ELOC* ($Q = 0.001$), and *PBRM1* and *SETD2* ($Q = 1.33 \times 10^{-5}$), tended to co-occur (Supplementary Data 5). In five tumours, *BAP1* mutations were the sole driver.

To search for non-coding drivers in gene promoters, untranslated and non-canonical splice regions, we used OncodriveFML[33], ActiveDriverWGS[34], and negative binomial regression adjusting for trinucleotide mutational context[35]. By collating and post processing of these data we nominated nine significant non-coding elements as potential candidate cancer drivers (Supplementary Methods).

The *TERT* promoter associations were primarily driven by the canonical mutations 5:1295113G > A, and 5:1295135G > A which have also been documented in TRACERx[36] and are early drivers in bladder cancer[37–39]. The 5′UTR of *TERT* was also recurrently mutated with 5:1295046T > G, which has previously been implicated in glioma and bladder cancer accounting for 23% of mutations within the region[39,40].

Our analysis also implicates the distal promoter of the *BRD2* gene, which together with other bromodomain proteins plays a role in histone acetylation modification in RCC. Other potential non-coding drivers included the distal promoter regions of *RSU1, FANK1*, and *BCAT2*[41,42] (Supplementary Data 6).

Systematic analyses of cancer genomes provide an opportunity of estimating the number of patients eligible for a targeted therapy and identify opportunities for therapeutic interventions. We assessed the clinical actionability of driver gene mutations by referencing OncoKB Knowledge Base[43] (version 3.11), and found 60 unique alterations were

targetable (OncoKB Level 1–4), and were all at least Level 4 (compelling biological evidence supporting the biomarker being predictive of drug response). We also examined the COSMIC Mutation Actionability in Precision Oncology[44] database highlighting an additional 708 unique alterations which are potentially targetable (Supplementary Fig. 5a, b, Supplementary Data 7).

## Recurrent structural and copy number alterations

In addition to the previously reported common 3p loss, 5q gain and 14q loss[7,36] we identified 25 other arm-level alterations that occurred more frequently than expected (Fig. 4a, Supplementary Data 8). We used GISTIC2[45] to identify genomic regions recurrently affected by focal amplifications and deletions ($Q < 0.05$; Fig. 4a, Supplementary Data 9). Aside from the previously reported CNAs[7], including del9p21.3 (*CDKN2A*), del3p12.2 (*GBE1*), amp5q35.3 (*SQSTM1*) and amp 8q24.21 (*MYC*) we identified four candidate CNAs: amp2q31.1 (*ACVR2A, CASP8, NFE2L2, PMS1, SF3B1*), amp13q34 (*ERCC5*), amp12p11.21 and del22q11.23. Of the genes implicated by these candidate focal amplifications, *NFE2L2* and *SF3B1* are documented to be oncogenic[46–49]. 16.6% of tumours showed whole genome duplication (WGD), a finding almost identical to the 15% reported by TracerX[50] (Supplementary Data 2). Complex chromosomal rearrangements were a feature of 60% of tumours with two-thirds of these displaying hallmarks of chromothripsis. As previously documented[7,36] the most frequent pattern on 3p loss was from rearrangement between 3p and 5q, ascribable to chromothripsis.

We identified 37 hotspots of recurrent simple SVs (FDR < 0.05) by piecewise constant fitting adjusting for local genomic features known to influence rearrangement density (chromatin accessibility, repeated elements, GC content, replication timing, gene density and expression). Fragile sites are prone to rearrangement (possibly due to replication error) and tend to co-occur with large, late-replicating genes. SVs occurring at fragile sites are hence likely to be the consequence of mechanistic rather than selective factors. After excluding 14 SV hotspots mapping to potential fragile sites, we identified 23 SV hotspots (Fig. 4b, Supplementary Figs. 6a–t, 7a–l, 8a–e, Supplementary Data 10). We identified a total of 66 breakpoints within 5p15.33, spanning *TERT*. These included, a deletion breakpoint and an unclassified event 2 kb downstream of the *TERT* promoter, and tandem duplications overlapping *TERT* ($n = 5$). In tumours from the 34 patients with a *TERT* 5′ UTR mutations, there were no overlapping unclassified/tandem-duplication events or a SV deletion/unclassified promoter breakpoints; an observation consistent with earlier findings[36].

## Mutational signatures

To gain insight into mutational processes in ccRCC, we extracted single-base substitution (SBS), double-base-substitution (DBS) and indel (ID) signatures de novo and related those to known COSMIC signatures (v3.2) using SigProfilerExtractor[51,52]. In the majority of cancers, single base substitutions could be assigned to signatures SBS5/SBS40 and SBS1 (nomenclature as per COSMIC) resulting from clock-like mutagenic processes. Other signatures recovered with known specific underlying aetiology include those associated with oxidative damage (SBS18), defective base excision repair (SBS30), APOBEC (SBS2, SBS13), tobacco smoking (SBS4, DBS2, ID3) and aristolochic acid (SBS22) (Fig. 5, Supplementary Fig. 9a, b, Supplementary Data 11). Three of the 7 tumours with SBS2 or SBS13 harboured a somatic mutation in an APOBEC gene, however, none of them are pathogenic. While the incidence of renal cancer has been linked to aristolochic acid exposure in residents of Danube river countries[53], 88% of patients with SBS22 tumour activity in the Gel cohort were self-reported to be white British. SBS31 and SBS35 have been attributable to platinum chemotherapy. We recovered SBS35 in four cases, but none had a reported past history of platinum chemotherapy. In contrast the tumours from five patients, which had a past history of a non-RCC cancer and had

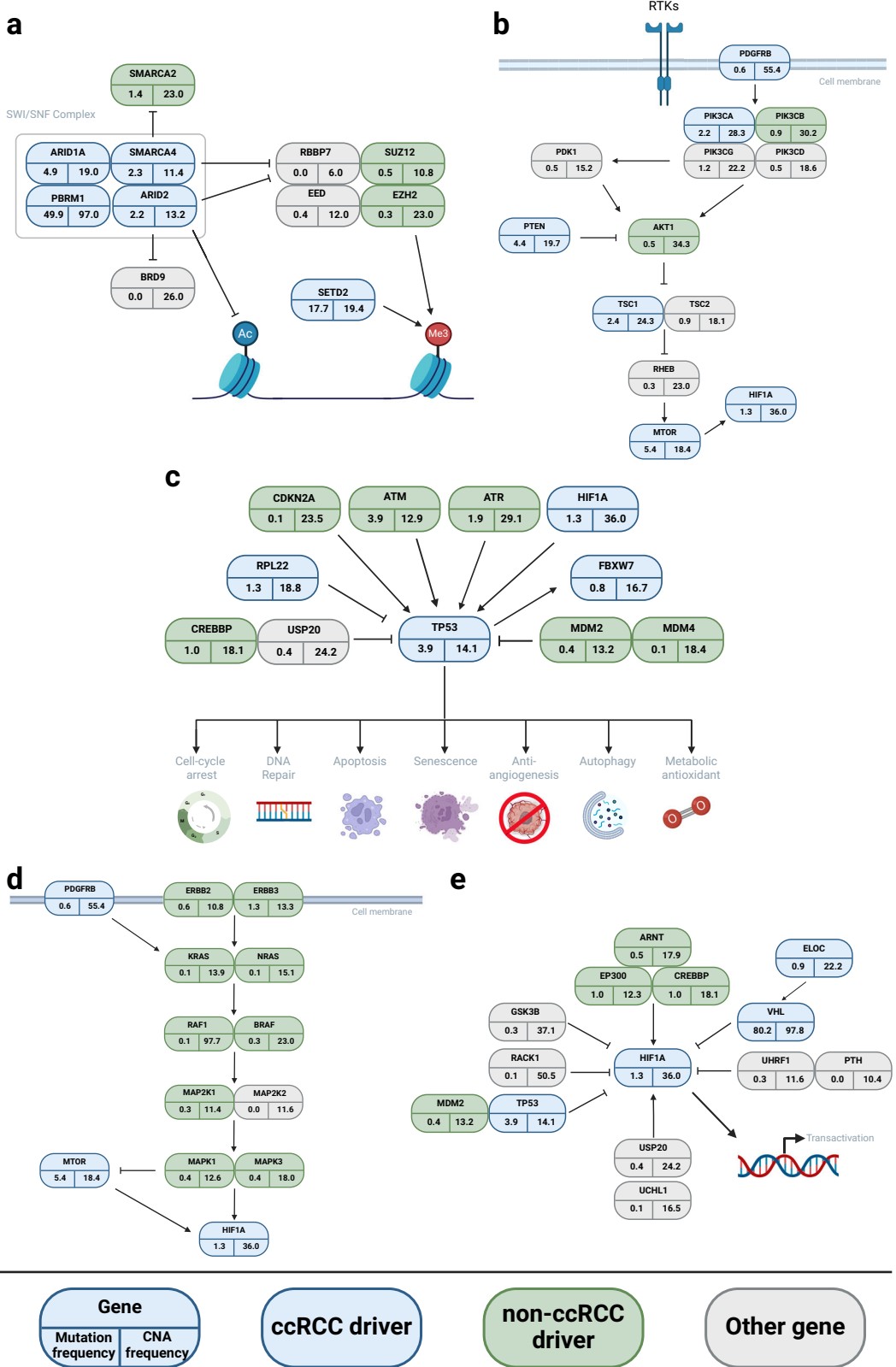

**Fig. 3 | Biological pathways in ccRCC. a** The *SWI/SNF* pathway. **b** The *PI3K/AKT/MTOR* signalling pathway. **c** The *TP53* pathway. **d** The *RAS/ERK* pathway. **e** The *VHL/HIF1A* and hypoxia pathway. Driver genes identified shown in blue, non-ccRCC driver genes in green and other pathway genes in grey. Non-ccRCC driver genes are defined as those identified in any other cancer. The number in the bottom left is the nonsynonymous mutational frequency and the number in the bottom right the copy number alteration (CNA) frequency. Figure 3 created with BioRender.com released under a Creative Commons Attribution-NonCommercial-NoDerivs 4.0 International license.

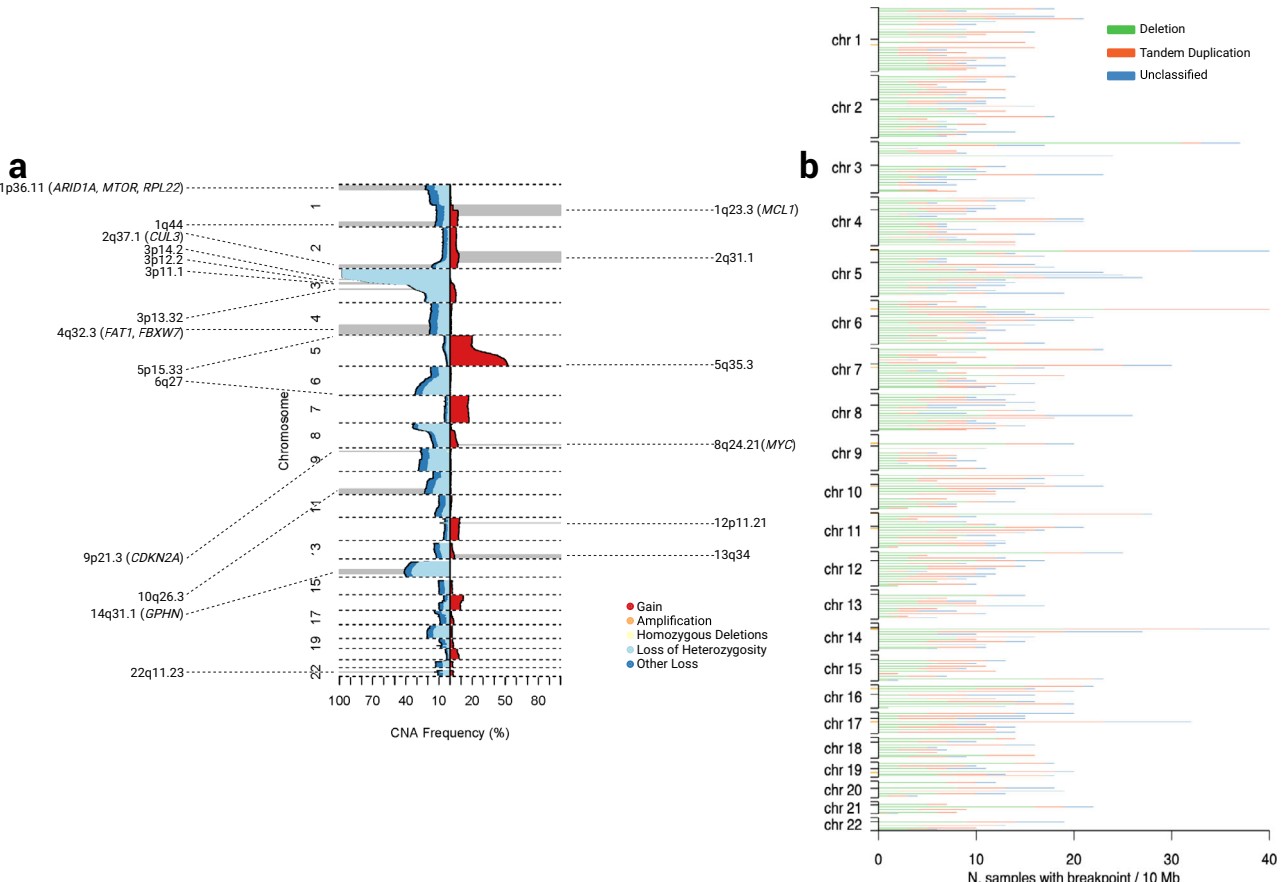

**Fig. 4 | Copy number alterations and structural variants. a** Frequency of copy number alterations across the ccRCC cohort. Copy number losses are coded in blue shades and copy number gains shown in red. The focal copy number alterations, as identified by GISTIC, are annotated along with predicted target gene. **b** The distribution of structural variants across the ccRCC cohort. Structural variants are classified as deletions, tandem duplications or unclassified structural variants. The black ticks on the y axis correspond to the chromosome start, centromere and end position while the orange ticks represent identified structural variant hotspot regions.

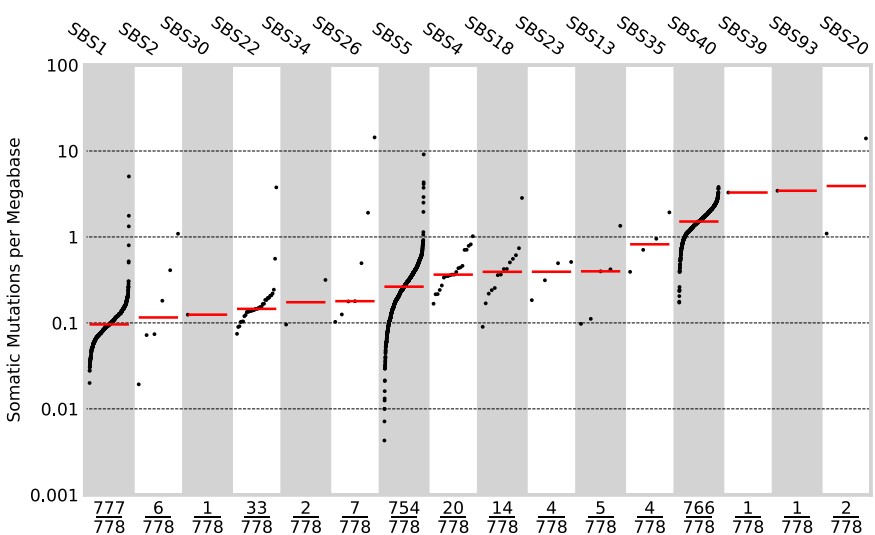

**Fig. 5 | Mutational signatures.** The mutational burden of single base substitution signatures (*n* = 778), as defined in COSMIC. The x axis indicates the number of samples for which each signature was active. The y axis is the mutational burden of the given signature. SBS1, SBS5 and SBS40, present in the majority of samples, are clock-like signatures. The higher average mutational burden of SBS40 signifies that it is the predominant signature active in the majority of samples.

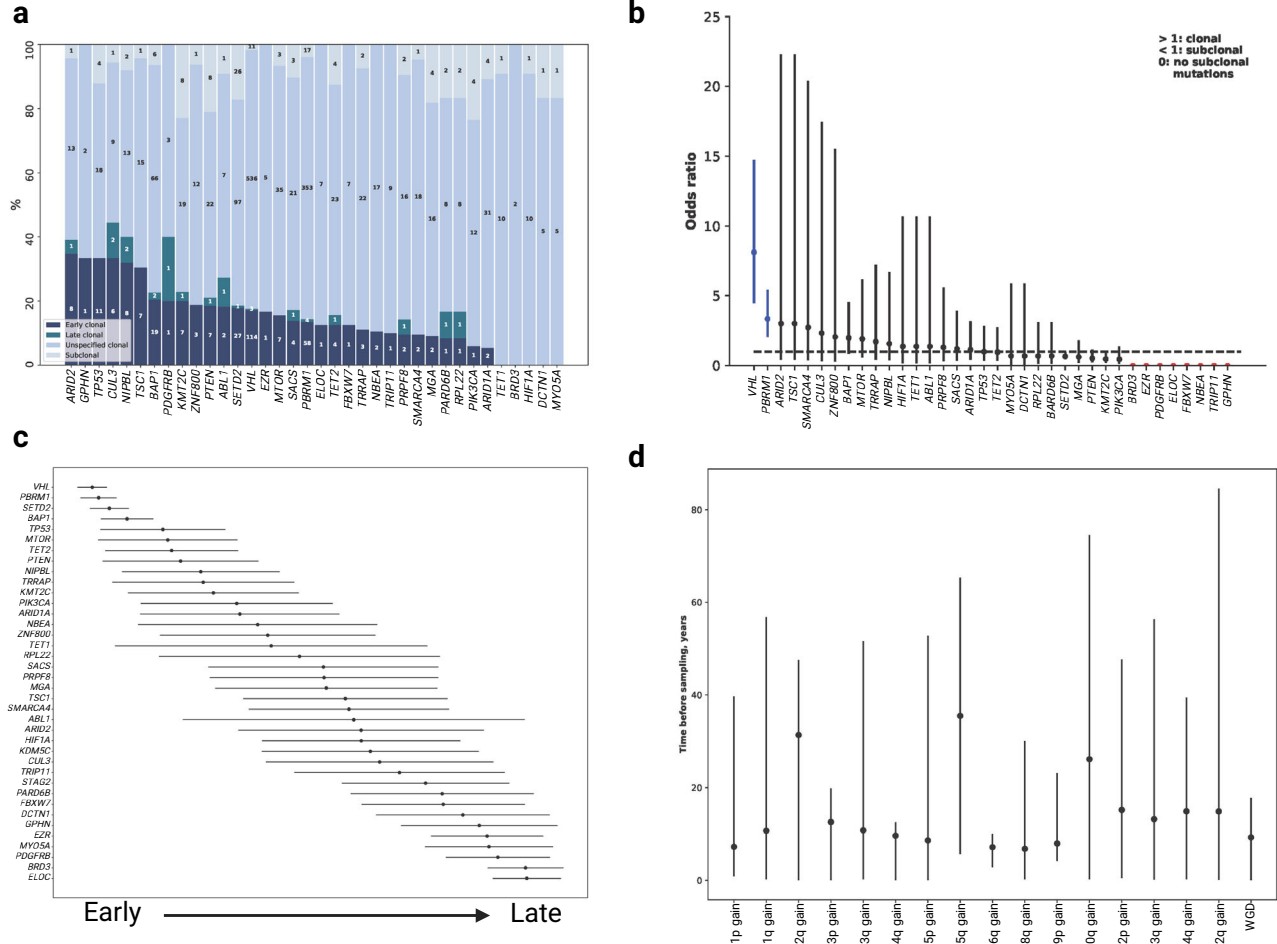

**Fig. 6 | Mutational timing. a** The proportion of clonal and subclonal nonsynonymous mutations (*n* = 775) in driver genes. **b** Odds ratio (OR) with 95% confidence intervals that a mutation in a driver gene is clonal relative to mutations in the other driver genes; OR > 1.0 indicates a mutation is more likely to be clonal. The genes in blue are significantly more likely to be clonal or subclonal. Genes in red have no subclonal mutations. **c** The relative ordering of nonsynonymous mutations in driver genes. The error bars correspond to the 95% confidence intervals of the average finishing position in the league model. The ordering was determined based on the distributions in (**a**) (Supplementary Methods). **d** Estimates, with 95% confidence intervals, of the real time at which copy number gains occur during tumour evolution.

received carboplatin or oxaliplatin did not display SBS31 or SBS35 (Fig. 5, Supplementary Data 11). To complement SigProfilerExtractor we searched for mutational signatures associated with defective mismatch repair (dMMR) and defective homologous recombination (dHR) using mSINGs[54] and HRdetect[55]. Two of the 3 tumours with evidence of dMMR (2 with SBS20, SBS26 and hypermutated phenotype) harboured *MLH1* somatic mutations which were accompanied by LOH, while the other carried a *POLE* mutation of unknown significance. None of these 3 cases carried germline pathogenic MMR variants and during the period of the study none developed metastatic RCC or were treated with ICIs. No case showed evidence of dHR. Considering mutational signature activity between clonal and subclonal mutations we found no significant enrichment or depletion of any SBS signatures between clonal and subclonal mutations.

### Ordering of mutational events

Using PhylogicNDT[56] in conjunction with MutationTimeR[57], we reconstructed the chronological ordering of focal CNAs and driver mutations. These methods estimate the time at which copy number gains occur by considering the fraction of mutations before and after the gain. In the regions of copy number gain individual mutations can therefore be estimated to be early or late events; subclonal mutations will, by definition, have occurred late in tumourigensis. Across all

tumours gain of 5q were consistently earlier alterations. As expected, mutations in *VHL*, *PBRM1*, *SETD2* and *BAP1* were predicted to be early events, with higher odds of harbouring clonal mutations, generally occurring before corresponding CNAs. In contrast, mutations in *KDM5C*, *STAG2* and *BRD3* were late events (Fig. 6a–c). Estimating the chronological timing of CNAs under varying mutational rates and tumour initiation time (Supplementary Methods) implies WGD occurred on average 9.2 years before tumour sampling and gain of 5q, 35.5 years before sampling (Fig. 6d). Moreover, the estimated lead time of 5q gain and WGD were both correlated with age at presentation (adjusting for grade and stage $P = 9.4 \times 10^{-13}$ and 0.02, respectively).

### Immune profile and evasion

Increased T-cell receptor alpha (TCRA) T-cell fraction[58] (i.e. fraction of T-cells present in the WGS sample), reflecting immune infiltration, was associated with increased tumour grade ($P = 4.0 \times 10^{-4}$; Supplementary Data 12). Considering TCRA T-cell fraction as a proxy for CD8+ infiltration, higher immune cell infiltration has previously been reported in a subset of inflamed *BAP1* or *PBRM1* mutated ccRCC tumours[59]. Consistent with this observation in our study *BAP1* mutated tumours were associated with higher TCRA T-cell fraction (OR = 3.43, 95% CI: 1.91–6.15), but no similar relationship was shown for PBRM1 (OR = 0.73, 95% CI: 0.53–1.01). We also noted that TCRA T-cell fraction was lower in

*MTOR*-mutated tumours (OR = 0.44, 95% CI: 0.2–0.97; Supplementary Data 13).

Using pVAC-Seq[60], we predicted 24,893 class I neoantigens across the 778 tumours (1–327 per tumour, median 26), resulting from: 66.5% missense mutations, 32.0% frameshift variants, 1.3% inframe deletions and 0.25% inframe insertions (Fig. 7a). Neoantigen burden was not correlated with stage or tumour grade (including presence of sarcomatoid features) but was positively associated with TMB (OR = 1.21, 95% CI: 1.18–1.23; Supplementary Data 12 and 14).

Examining evidence of immune evasion we LOH or mutation of HLA class I genes (*HLA-A*, *HLA-B*, *HLA-C*) and immune escape genes (Supplementary Methods). Using LOHHLA, we detected LOH of HLA in only 5.9% of tumours. It has been reported that LOH on HLA class I genes and 9p21.3 loss tend to co-occur[61], suggesting a potential mechanism for immune escape. After adjusting for stage and grade we found no evidence to support such a relationship (OR = 1.32, 95% CI: 0.58–2.99; Supplementary Data 15). We also found no association between indel burden and HLA allele status (OR = 0.92, 95% CI: 0.82–1.03; Supplementary Data 16). Nonsynonymous mutations of HLA genes were rare (0.5%). An inactivating mutation in at least one of the 22 antigen presenting genes[62,63] (APG) was seen in only 3.1% (24/778) of tumours (Fig. 7b). None of the APGs displayed a propensity for mutation. Collectively, on the basis of alteration of these escape pathways (HLA LOH, APG inactivation or HLA somatic mutation) only 9.0% (70/778) of tumours were predicted to exhibit some form of genetically-driven immune evasion.

### Clinico-pathological relationships

Increased tumour grade was associated with necrosis ($P = 9.4 \times 10^{-15}$), increased TMB ($P = 0.001$) and mutational sub-clonality ($P = 0.02$). High SV count ($P = 1.5 \times 10^{-11}$), WGD ($P = 1.2 \times 10^{-6}$) and weighted genome instability index ($P = 1.7 \times 10^{-11}$) were associated with higher tumour grade. Consistent with previous literature[64], tumours with mutations in *BAP1* and *TP53* were more likely to present as high grade ($P = 1.8 \times 10^{-5}$, $P = 2.1 \times 10^{-6}$, respectively; Supplementary Data 12). The genomic landscape of several cancers have been shown to differ by age at diagnosis[65]. To explore this possibility with respect to ccRCC we compared cases diagnosed younger than age 46 and older. This stratification provided no evidence for significant differences in genomic features other than TMB (Supplementary Data 12).

Information was available on 605 patients for us to examine the relationship between genomic features with overall survival (OS) and cancer specific survival (CSS) (Supplementary Figs. 10a–d, 11, 12a–f, 13, Supplementary Data 1, 17, 18). Tumour grade and stage and sarcomatoid features were strong predictors of OS and CSS (Supplementary Data 17 and 18). After adjusting for co-variants using Cox regression increased OS and CSS was associated with *VHL* (Hazard Ratio (HR) = 0.60, 95% CI: 0.36–0.98; HR = 0.60, 95% CI: 0.3–1.19) and *PBRM1* (HR = 0.64, 95% CI: 0.42–0.97; HR = 0.55, 95% CI: 0.32–0.95 respectively) mutation status. Given the co-occurrence of *VHL* and *PBRM1* mutations (84%, 325/388 of *PBRM1*-positive tumours were also *VHL* mutated), after adjusting for *VHL* status, the association of *PBRM1* mutational status was no longer statistically significant for either OS or CSS (HR = 0.68, 95% CI: 0.44–1.03; HR = 0.58, 95% CI = 0.33–1.02). Aside from *VHL*, mutation of no other driver gene showed an independent association with patient survival; acknowledging we had limited statistical power to demonstrate a relationship with less frequently mutated genes (Fig. 8a–e, Supplementary Fig. 11). After adjusting for *VHL* status, higher SV count was, however, associated with worse OS (HR = 1.01, 95% CI: 1.00–1.10) and CSS (HR = 1.01, 95% CI: 1.01–1.02; Fig. 8c, Supplementary Figs. 11, 12e, 13, Supplementary Data 18).

While we found no association between OS or CSS and either neoantigen burden or immune escape, a higher TCRA T-cell fraction was associated with a better OS (HR = 0.65, 95% CI: 0.43–0.99; Fig. 8d).

Aside from an association between DBS4 and CSS, we found no other significant associations between additional mutational signatures as independent predictors of either OS or CSS. We also did not find evidence to support a relationship between either OS or CSS with intratumour heterogeneity or wGII, both features which have previously been purported to influence prognosis[25,66] (Supplementary Figs. 11 and 13, Supplementary Data 18).

We were able to examine the relationship between molecular features and progression-free survival (PFS) in 167 of the patients ascertained on the basis of being at intermediate-high risk of tumour recurrence on the basis of their Leibovich score[67] (Supplementary Data 1). While *VHL* status was not associated with better PFS we observed that *KDM5C* mutation was independently associated with worse outcome (HR = 1.98, 95% CI: 1.00–3.91; Supplementary Figs. 14a–h and 15, Supplementary Data 19) and a higher incidence of necrosis (OR = 4.81, 95% CI: 1.20–19.11; Supplementary Data 20). Thirty-seven of the 167 patients had received ICI therapy as a first or second line treatment and in 21 of these there was documented evidence of clinical benefit. Restricting our analysis to these 37 patients, we found no evidence of a difference in ICI response in patients with sarcomatoid tumours features and only deletion of 6q was associated with clinical benefit (OR = 7.66, 95% CI: 1.11–52.55). Although not statistically significant, but consistent with other reports[68], sarcomatoid histology was associated with better response to ICI (Supplementary Data 21).

## Discussion

This study represents a comprehensive description of the genomic landscape of ccRCC by utilising a large cohort of WGS samples. We acknowledge that there are limitations to our analysis. Specifically, our reliance on short-read sequencing and lack of transcriptomic information. During the period of this study the classification of the renal tumours was conducted in accordance with the WHO Classification of Urinary and Male Genital Tumours (4th and 3rd Editions). Although there was no change to the diagnostic criteria for ccRCC between WHO 3 and 4, the classification of RCC has evolved with new entities recognised within the (current) WHO Classification (5th Ed, 2022) that were not previously 'recognised'/accepted entities. Some of these mimic ccRCC and there is therefore a chance that a small number of cases within our study cohort might be re-classified if these were subject to contemporary review. Such considerations will also pertain to previously published studies conducted by TCGA and PCAWG. Accepting the caveats, as well as confirming established ccRCC driver genes we identify candidate drivers further highlighting oncogenic metabolism and epigenetic reprogramming as being central to ccRCC biology. Additionally, we confirm p*TERT* mutations as drivers, thereby further substantiating telomerase dysfunction in the development of ccRCC. Mutational signature analysis provides a mechanistic basis for known lifestyle and exposure risk factors as well as potentially indirectly suggesting additional ones. While we did not identify any new mutational signatures, our analysis provides further support for tobacco smoking being a risk factor for ccRCC[69].

The large size of our study, coupled with the standardised management protocols for ccRCC patients within the UK National Health System, has enabled us to investigate the correlation between molecular features and patient prognosis. The clinical course for many ccRCC patients with apparent same stage disease can be highly variable. Upfront identification of patients who are likely to relapse early offers the prospect of intervening preemptively to maintain remission. Furthermore, since metastatic ccRCCs are chemotherapy and radiotherapy resistant, identifying tumour sub-groups with targetable molecular dependencies has the potential to inform on biologically driven therapies.

The relationship between mutations in the major clonal driver genes and patient survival has been the subject of a number of previous studies, but findings have been inconsistent[6,7,70–76]

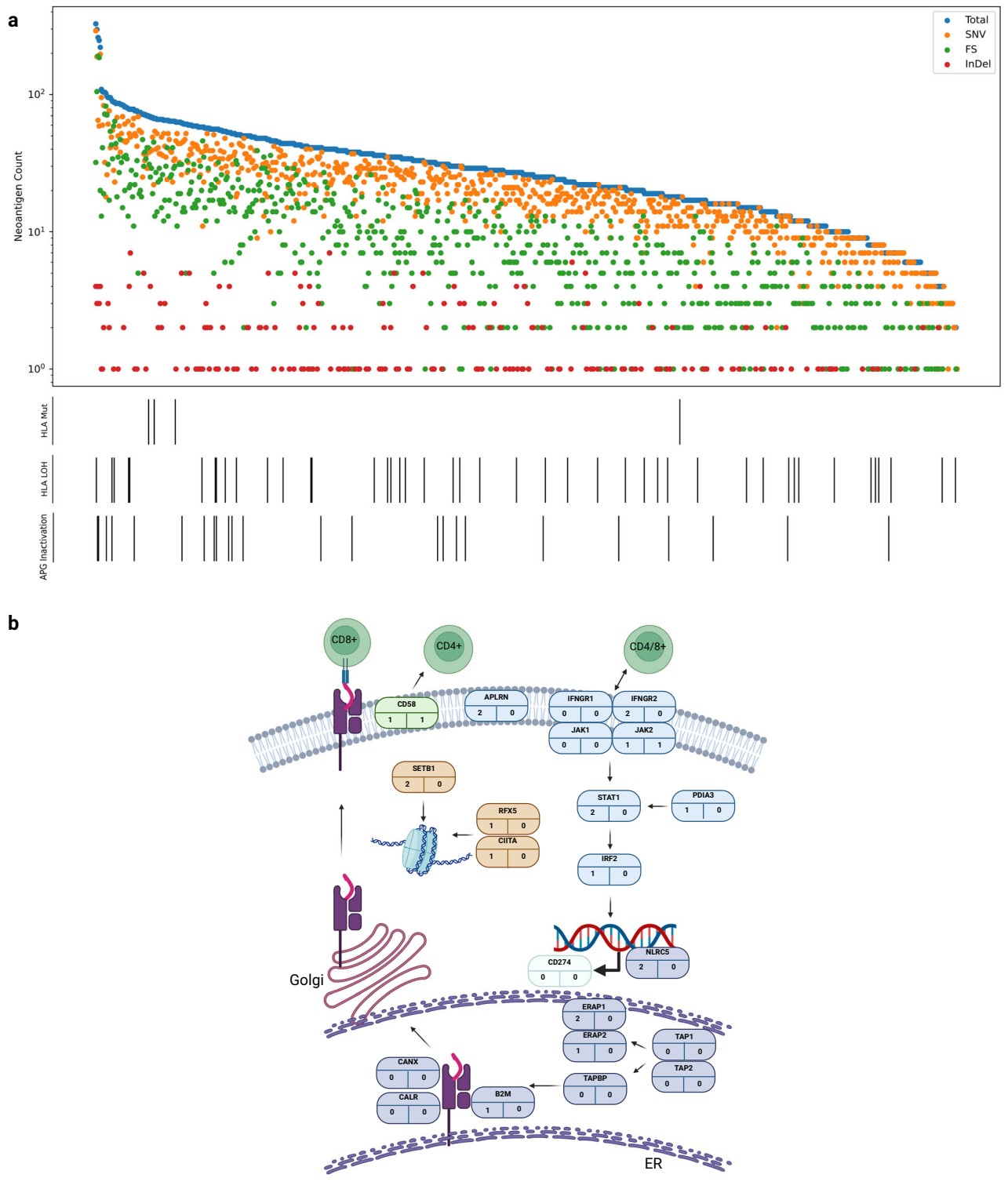

**Fig. 7 | Immune landscape of ccRCC. a** Neoantigen burden and immune escape mutations. Lower bars show antigen processing genes (APG) and human leucocyte antigens (HLA) alterations present in each cancer. **b** Somatic mutations in each of the antigen presentation pathway genes. The number in the bottom left is the truncating mutation count and the number in the bottom right is the number of biallelic nonsynonymous mutations. APGs in purple, IFN-γ pathway genes in blue, epigenetic modifier genes in brown, CD274 comprises the PD-L1 receptor and CD58 receptor is encoded by CD58. Figure 7b created with BioRender.com released under a Creative Commons Attribution-NonCommercial-NoDerivs 4.0 International license.

(Supplementary Data 22). While some studies[7,9] have reported *BAP1* mutations being associated with a worse clinical outcome, other studies[71,75] have failed to demonstrate any relationship. As previously documented[9,77], and herein, *BAP1* mutations are strongly associated with increased grade and after adjustment we failed to show support for an independent relationship. In our study, we, however, show *VHL* mutation status was independently associated with an improved OS, consistent with a recent study[74]. *VHL* mutations are early events of ccRCC development whereas other mutated genes are acquired later therefore they might be assumed to play more of a role in disease

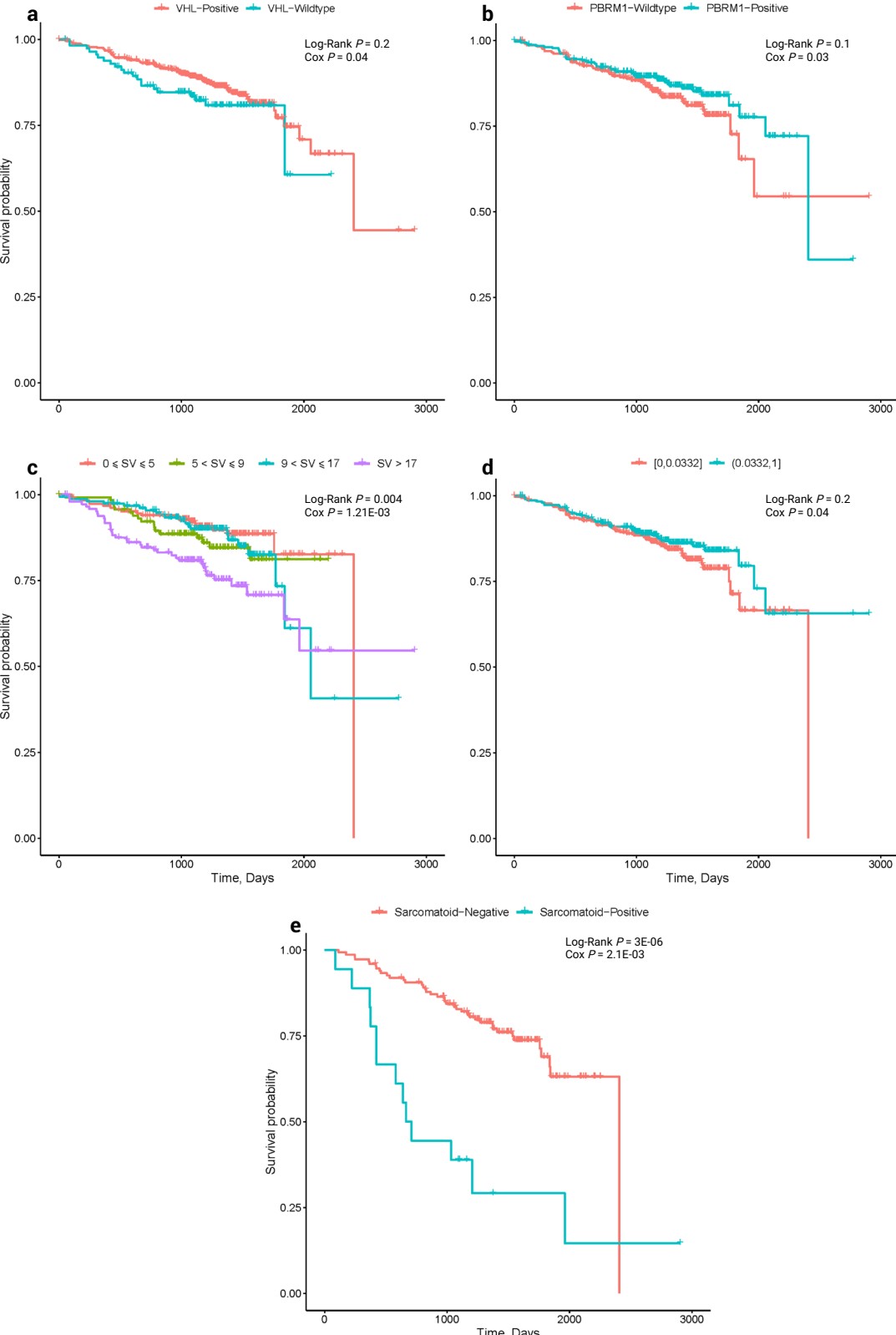

**Fig. 8 | Kaplan–Meier survival curves of overall survival (OS).** Relationship between OS and **a** VHL mutation status ($n = 605$); **b** PBRM1 mutation status ($n = 605$); **c** structural variant count ($n = 605$); (**d**) TCRA T-cell fraction ($n = 605$); **e** sarcomatoid ($n = 167$). Log-Rank $P$ and Cox $P$ refers to the Log Rank test and Cox Regression (two-sided z-test), respectively.

progression. Hence it is unclear why *VHL*-positive ccRCC tumours might have a more favourable outcome than *VHL*-wildtype ccRCC. Distinct evolutionary subtypes of ccRCC have, however, been proposed that appear biologically and clinically distinct, with subtypes defined being by *VHL*-wildtype, *VHL*-monodrivers, and those with multiple clonal drivers[8]. After adjusting for *VHL* status, we did not detect a statistically significant independent association between other driver mutations and survival. Amongst the strongest

relationships we identified was between increased copy number with increased survival, which was independent of tumour grade, presumably reflecting tumour heterogeneity. We did not find support for the purported relationship between intra-tumour heterogeneity and prognosis[25], however, our analysis did not benefit from multi-region sampling.

Although current drug treatment paradigms for ccRCC exploit targeted therapies they are primarily not directed against any specific genomic feature. To investigate the prospect of targeting specific driver mutations we queried OncoKB[43], which is regularly curated by an expert panel and therefore generally considered to reflect the current state of knowledge. Since other investigators have reported a higher targetable variant detection rate by applying multiple tools to annotate variants we also made use of The COSMIC Mutation Actionability in Precision Oncology resource[44]. The majority of the alterations we describe as being actionable are based on clinical evidence from other cancers or biological plausibility. As per previous reports, the majority of the targetable alterations we identified are within PI3K/mTOR pathway genes. Randomised clinical trials showing clinical benefit of the mTOR inhibitors temsirolimus and everolimus in RCC have already led to their regulatory agency approval. Other targets have not been specifically studied in the context of ccRCC, hence results cannot be interpreted as definitive proof of response prediction. Examples of drugs where the population of interest could be expanded to treat ccRCC include: Temsirolimus, which is undergoing ongoing trials as a treatment for *FBXW7*-positive solid tumours[78,79], nilotinib for *ABL1* mutations[80], niraparib for *BAP1* mutations[81], Tazemetostat hydrobromide for *SMARCA4* mutated cancers[82,83], olaparib with pembrolizumab for *ARID2*-positive melanoma[84], and alpelisib for *PIK3CA* in ER-positive metastatic breast cancer[85,86]. An important caveat to our analysis is that the genetic profiles we derived are of a single region, which has potentially limited our ability to detect clinically important subclonal targetable alterations.

In many other cancers, a high mutational and neoantigen burden have been linked to better overall survival and responsiveness to checkpoint inhibitors presumably reflecting native immune responsiveness[87]. In our study, there was no association between neoantigen burden and OS. In contrast, there was a strong relationship between increased T-cell infiltration and better prognosis. While this might seem counterintuitive, however, this finding may be explained by the poor accuracy (6%) of current HLA-affinity-based neoantigen prediction algorithms[88]. Accepting these limitations, the twin observations of higher T-cell infiltration being associated with better outcome and genetically predicted immune evasion is uncommon and supports the rationale for immunotherapy.

There is interest in the prospect of population screening for RCC, given the rising incidence of the disease, the high proportion of asymptomatic individuals at diagnosis and associated high mortality rate. Our analysis supports previous work suggesting that ccRCC driver mutations often precede diagnosis by many years, if not decades[36], information relevant to the design of any screening programme.

Although some cancers have reaped demonstrable benefits from the current genomic revolution, the same benefits have not been yet observed in RCC, and further efforts should be directed to identify the precise role of genomic tumour profiling in the clinical setting.

## Methods
### Ethics statement
This study was conducted as part of the 100,000 Genomes Projects and approved by the East of England – Cambridge South Research Ethics Committee (REC reference: 14/EE/1112). All patients provided written informed consent and the study was conducted in accordance with the Declaration of Helsinki (see https://www.genomicsengland.co.uk/initiatives/100000-genomes-project/documentation for further information on patient consent and withdrawal). Sex-stratified analysis was not considered for this investigation, and there are no results that are only applicable to a single sex.

### The 100kGP cohort
The analysed cohort comprised tumour-normal sample pairs from patients with primary RCC recruited to 100kGP (v14 release) through 13 Genomic Medicine Centres across England[10,11] (Fig. 1). The renal tumour cases included in this study were all routine surgical cases reported by diagnostic histopathologists at contributing centres. Histology of RCC was as per WHO Classification of Urinary and Male Genital tumours 3rd/4th edition[89], which have matching clear cell renal cell carcinoma (ccRCC) diagnostic criteria. The collection of tissue and the preparation, extraction and quantification of DNA was undertaken locally, followed by transfer of DNA to a central biorepository. We restricted our analysis to high-quality sequencing data derived from PCR-free, flash-frozen primary tumour samples from 10,470 adults (Supplementary Data 23). Illumina conducted whole genome sequencing of paired tumour/normal DNA. Processed BAM files were sent to Genomics England, who performed additional quality checks and managed data storage. We corrected for reference bias in calling of variants using FixVAF[12]. We selected one tumour/normal pairing per patient. We used Strelka to call somatic variants[90], a four-stage pipeline incorporating Battenberg[13] for copy number calling (Supplementary Data 25) and a consensus approach based on Delly[16], Lumpy[15] and Manta[14] for calling somatic structural variants (SVs). To assess the clinical relevance of individual mutations, in addition to Ensembl Variant Effect Prediction[91], all unique missense SNVs were annotated using AlphaMissense[24] to compare pathogenicity between driver and non-driver mutations. Comprehensive clinico-pathology information on the patients is provided in Supplementary Data 1. Complete details on sample curation, tumour purity estimation, WGS, somatic variant calling, mutation annotation, copy number alteration calling/annotation, somatic structural variant calling/annotation, whole genome duplication annotation and tumour/germline telomere length estimation provided in "Supplementary Methods".

### Identification of drivers and driver annotation
We used IntOGen to identify coding drivers[18] (Supplementary Methods). The relative evolutionary timings of candidate driver mutations were obtained using MutationTimeR (Supplementary Methods). We considered gene perturbation screening data from DepMap to determine the functional basis of candidate driver genes[30,31]. To complement this approach, we also considered gene expression data from TCGA[28] and GTEx[29], accessed through GEPIA[92]. We assessed the clinical actionability of driver gene mutations by interrogating OncoKB Knowledge Base[43] (version 3.11) and the COSMIC Mutation Actionability in Precision Oncology database[44]. We searched for non-coding drivers within core promoters, distal promoters, 5′ and 3′ UTRs of canonical protein-coding transcripts, non-canonical splice regions, and LincRNAs using OncodriveFML[33], ActiveDriverWGS[34] and Negative binomial regression modelling[35]. We used Empirical Brown's method to combine *P*-values from methods and adjusted for multiple-testing using the Benjamini-Hochberg procedure (Supplementary Methods).

Cellular pathways containing driver genes (Supplementary Data 4), identified from both the IntoGen and non-coding pipelines involved in tumourigenesis, were referenced to the literature using PubMed. Pathways were also interrogated using ActivePathways[93] and MSigDB[94,95] (v7.5.1).

### Copy number and structural variant hotspots
Recurrent arm-level copy number events, focal amplifications and deletions, were identified using Genomic Identification of Significant Targets in Cancer[45] (GISTIC, v2.0.2.3). SVs were classified into simple and complex SVs using ClusterSV[96] (Supplementary Methods) and

structural variation hotspots for deletion, tandem duplication and simple unclassified SVs were identified using a permutation-based approach as per as Glodzik et al.[97]. Additional filtering was applied to identify if the hotspot regions were predicted fragile regions in the genome (Supplementary Methods).

### Mutational signatures

De novo extraction of single-base-substitution (SBS), doublet-base-substitution (DBS) and insertion and deletion (ID) signatures, including decomposition to known COSMIC signatures[44] (v3.2), was performed using SigProfilerExtractor[51]. We complemented this using mSINGS[54] and HRDetect[55,98] to specifically identify mismatch repair deficient and homologous recombination deficient tumours respectively (Supplementary Methods).

### Immune escape

All genomes were HLA-typed using POLYmorphic loci reSOLVER[99] (POLYSOLVER). Neoantigens were predicted using personalised Variant Antigens by Cancer Sequencing[60] (pVAC-Seq). We classed tumours as exhibiting immune escape on the basis of a non-synonymous mutation, or loss-of-heterozygosity in any one of three HLA Class-I genes, or an inactivating mutation in an antigen presenting gene (Supplementary Methods).

### Clinical correlations and survival analysis

Correlations between clinical and mutational properties were identified using logistic, linear and negative binomial regression. Both univariate and multivariate regression adjusting for sex, age of sampling, stage and grade was considered (Supplementary Methods). Overall survival and cancer-specific survival was defined as the time from the date of sampling to death from any cause. Kaplan–Meier survival curves were generated and the homogeneity between groups was evaluated with the log-rank test. Progression-free survival was defined as the time from the date of sampling to radiological progression. Cox regression analysis was used to estimate hazard ratios and respective 95% confidence intervals, and adjustment for clinical variables was performed by multivariable analysis.

### Reporting summary

Further information on research design is available in the Nature Portfolio Reporting Summary linked to this article.

## Data availability

The data supporting the findings of this study have been deposited in the National Genomic Research Library and can be accessed via the Genomics England Research Environment, a secure cloud workspace. The raw data, including patient profiles and corresponding genomic sequencing data, are only available under restricted access for patient privacy reasons. Access can be obtained by first applying to become a member of either the Genomics England Research Network (https://www.genomicsengland.co.uk/research/academic) or the Discovery Forum (industry partners https://www.genomicsengland.co.uk/research/research-environment). The process for joining the network is described at https://www.genomicsengland.co.uk/research/academic/join-gecip and consists of the following steps: (1) Your institution will need to sign a participation agreement available at https://files.genomicsengland.co.uk/documents/Genomics-England-GeCIP-Participation-Agreement-v2.0.pdf and email the signed version to gecip-help@genomicsengland.co.uk. (2) Once you have confirmed your institution is registered and have found a domain of interest, you can apply through the online form at https://www.genomicsengland.co.uk/research/academic/join-gecip where you can specify the reason for access and expected timeframe that you wish to have access. Once your Research Portal account is created you will be able to login and track your application. (3) Your application will be reviewed within 10 working days. (4) Your institution will validate your affiliation. (5) You will need to complete the online Information Governance training and will be granted access to the Research Environment within 2 days of passing the online training. The processed clinical and genomic data applied to the investigation are available in the Research Environment within the folder /re_gecip/shared_allGeCIPs/rculliford/ccRCC_landscape. At present, there is no proposed end date for data access within the research environment. All other public/private datasets used in the study, including corresponding download links and version numbers, can be found in Supplementary Data 24.

## Code availability

After completion of the instructions given by the Data availability statement, code to allow for reproducibility of results and figures are available in the research environment within the folder /re_gecip/shared_allGeCIPs/rculliford/ccRCC_landscape. Sources of each software package and externally downloaded data are shown in Supplementary Data 24.

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

## Acknowledgements

R.S.H. acknowledges grant support from Cancer Research UK (C1298/A8362), the Wellcome Trust (214388) and the Medical Research Council. S.T. is funded by Cancer Research UK (A29911); the Francis Crick Institute, which receives its core funding from Cancer Research UK (FC10988), the UK Medical Research Council (FC10988), and the Wellcome Trust (FC10988); the National Institute for Health Research (NIHR) Biomedical Research Centre at the Royal Marsden Hospital and Institute of Cancer Research (grant reference number A109), the Royal Marsden Cancer Charity, The Rosetrees Trust (reference A2204), Ventana Medical Systems Inc (references 10467 and 10530), the National Institute of Health (U01 CA247439) and Melanoma Research Alliance (reference 686061). and the US Department of Defense (award W81XWH-22-1-0652) and VHL Alliance. This research was made possible through access to data in the National Genomic Research Library, which is managed by Genomics England Limited (a wholly owned company of the Department of Health and Social Care). The National Genomic Research Library holds data provided by patients and collected by the NHS as part of their care and data collected as part of their participation in research. The National Genomic Research Library is funded by the National Institute for

Health Research and NHS England. The Wellcome Trust, Cancer Research UK and the Medical Research Council have also funded research infrastructure.

## Author contributions

A.J.C., D.C., Z.T., H.P., J.L. and K.L. processed clinical data. L.B. and D.C. reviewed histology; A.J.C. performed sequencing data quality control. D.C. and A.J.C. performed quality control of simple somatic mutations. A.J.C. called copy number alterations. A.J.C., R.C. called structural variants and identified recurrent events. B.K. and S.E.D.L. identified recurrent copy number alterations. R.C. and B.K. identified driver mutations. A.G., D.W., A.F. and C.M. extracted mutational signatures. S.E.D.L. carried out HLA estimation, neoantigen prediction and immune escape prediction. R.B. provided TCRA T Cell fractions. B.K., C.M. estimated telomere content. B.K. and C.M. inferred driver mutation clonality and timing. C.M., R.C. and S.E.D.L. correlated genomic and clinical features. R.C. carried out survival analysis. C.M., R.C., S.E.D.L. A.S. and B.K. performed clinical actionability analyses. R.S.H. and S.T. supervised the study. R.S.H., R.C., C.M., S.E.D.L., Z.T., H.P. and S.T. wrote the manuscript. All authors read and approved the final version.

## Competing interests

S.T. has received speaking fees from Roche, AstraZeneca, Novartis and Ipsen. S.T. has the following patents filed: Indel mutations as a therapeutic target and predictive biomarker PCTGB2018/051892 and PCTGB2018/051893 and Clear Cell Renal Cell Carcinoma Biomarkers P113326GB. None of the other authors have a financial or non-financial conflict of interest.

## Additional information

## The Renal Cancer Genomics England Consortium

Mehran Afshar[14], Oyeyemi Akala[15], Janet Brown[16], Guy Faust[15], Kate Fife[17], Victoria Foy[18], Styliani Germanou[14], Megan Giles[19], Charlotte Grieco[20], Simon Grummet[21], Ankit Jain[22], Anuradha Kanwar[15], Andrew Protheroe[23], Iwan Raza[23], Ahmed Rehan[16], Sarah Rudman[24], Joseph Santiapillai[25], Naveed Sarwar[26], Pavetha Seeva[23], Amy Strong[17], Maria Toki[24], Maxine Tran[25], Rippie Tutika[21], Tom Waddell[18] & Matthew Wheater[19]

[14]St George's University Hospitals NHS Foundation Trust, London, UK. [15]University Hospitals of Leicester NHS Trust, Leicester, UK. [16]Sheffield Teaching Hospitals NHS Foundation Trust, Sheffield, UK. [17]Cambridge University Hospitals NHS Foundation Trust, Cambridge, UK. [18]Christie NHS Foundation Trust, Manchester, UK. [19]University Hospital Southampton NHS Foundation Trust, Southampton, UK. [20]The Royal Marsden NHS Foundation Trust, London, UK. [21]University Hospitals Birmingham NHS Foundation Trust, Birmingham, UK. [22]The Royal Wolverhampton NHS Trust, Wolverhampton, UK. [23]Oxford University Hospitals NHS Foundation Trust, Oxford, UK. [24]Guy's and St Thomas' NHS Foundation Trust, London, UK. [25]Royal Free London NHS Foundation Trust, London, UK. [26]Imperial College Healthcare NHS Trust, London, UK.

