## [Peer Review File · Nature Communications]

Whole genome sequencing refines stratification and therapy of patients with clear cell renal cell carcinomaReviewers' Comments:

Reviewer #1:

Remarks to the Author:

Culliford et al. analyzed whole genome sequencing (WGS) data from 778 ccRCC patients from the UK Genomics England 100,000 Genomes Project. This study provides a comprehensive picture of the genomic changes in ccRCC tumors. Especially, this study revealed some novel cancer drivers for ccRCC, even though the mutation frequencies for these potential drivers are low. Additionally, they have found associations between various genomic characterizations of the ccRCC patients/tumors and the clinical outcomes. This is a nicely performed, large scale, and comprehensive genomic analyses of ccRCC that will serve as a valuable resource for the research community. However, the claimed novel drivers were not carefully examined informatically and have not been experimentally validated. Furthermore, clinical associations shown in the manuscript are marginally significant and need careful evaluation and validation. These two aspects need to be addressed by the authors to demonstrate the value of this study.

There are a few additional specific comments and concerns.

1. Although the value of these WGS data and analyses as references is clear, most of the findings are confirming similar ones from various prior WES studies of ccRCC.
2. The novel drivers are potentially interesting in illustrating the etiology and key molecular events driving the progression of ccRCC. However, it may not be easy to determine if these are truly important drivers for ccRCC due to the low frequency and the absence of any functional studies.
3. ccRCC drivers have long been established and the order of the events have also been described in several major studies. How do the claimed new drivers fit into the big picture of ccRCC oncogenesis?
4. To gain a complete understanding of ccRCC drivers, the authors can combine TCGA, CPTAC, and their own WGS-based ccRCC data for a comprehensive driver discovery.
5. I would also like to see more detailed analysis of structural events using WGS data. Have a clear map of structural events and the heterogeneity of such events in tumors would be important for realizing the value of the WGS data.
6. Have the authors discovered any non-coding events from WGS potentially associated with ccRCC pathogenesis? This will provide novelty to this study.
7. Have the authors considered generating some long reads WGS data to perform a better map of some of the breakpoints? This will provide novelty to this study.

Minor points:

1. "Twenty-two of the patients (2.8%) were carriers of pathogenic germline variants in one of the well-established RCC susceptibility genes and 10 (1.2%) were carriers of a variant in another cancer susceptibility gene (Supplementary Table 3)." The description of the genes involved should be more specific.
2. Line 192 – "TMB" should be defined at its first use (Tumor mutational Burden).
3. The claim of "only 9% of tumors were predicted to exhibit some form of genetically-driven immune evasion" needs stronger, clear evidence.
4. The ordering of mutational events section lacks details. Please expand.

Reviewer #2:

Remarks to the Author:

Culliford, et al. report results from whole genome sequencing of 778 ccRCC patients enrolled in the 100,000 Genomes Project in relation to Driver mutations, Recurrent structural and copy number alterations, Mutational signatures, Ordering of mutational events, Immune evasion and Clinico-pathological relationships. Reassuringly, driver mutations are consistent with what has been previously reported in ccRCC, and the authors report 13 events which either had not been reported previously or have frequencies <1% in landmark genomics studies of ccRCC. The authors confirm the importance of TERT promoter mutations in RCC, demonstrate the smoking mutational signature in a subset of cases,

and demonstrate that only a minority of cases harbor alterations that would predict for lack of response to checkpoint immunotherapy. Overall, this is a large and comprehensive data set that is of interest to the renal cell carcinoma community, though the findings largely confirm existing understanding of ccRCC biology rather than provide novel insights.

I have the following largely minor comments:

1. In Figure 2, the authors highlight in red mutations that had previously been reported in <1% of patients in other RCC cohorts, but are there gene mutations detected in this study that have not been previously reported in RCC at all?
2. In Figure 3, does "non-ccRCC" mean RCC of other histologies, or does it mean in cancers other than RCC? The abbreviation could be changed, or this could be clarified in the legend.
3. Figure 5 would benefit from a figure legend. I am assuming that the numbers on the X-axis refer to the number of samples that have evidence of each signature, and that the Y-axis refers to somatic mutations per Mb in those samples, but it would help the reader to 1) understand what these signatures are (possibly with a Table below the figure) and 2) what it represents that certain signatures are associated with higher or lower somatic mutations/Mb.
4. The authors report "three tumours displayed a hypermutated phenotype: i.e., excessively high SNV/indel mutation burden 83 (maximal SNV/Mb = 33.65, maximal indel/Mb = 21.77)." and also note that "Three cases displayed evidence of dMMR, of which two harboured MLH1 somatic mutations accompanied by LOH, but none carried germline pathogenic MMR variants. Two of the cases had mutations assigned to signatures associated with dMMR (SBS20, SBS26)." Was there overlap between the hypermutated cases and those with dMMR? If so, this should be stated, and if not, why the discrepancy?
5. In Figure 6c, please clarify what the x-axis of Log(OR) represents. There seems to be a discrepancy in that TP53 mutations are more likely to be subclonal (with OR < 1 in Figure 6B) but are early events (Figure 6C) - can the authors comment on this?
6. In regards to the immune landscape, it would be interesting to know if ccRCC with sarcomatoid features has a different landscape than those without since sarcomatoid RCCs have more clinical benefit from immune checkpoint inhibitors. Is the presence of sarcomatoid features known in these cases?
7. In Figure 8, the Log-rank P value for OS for VHL mutation was 0.2 and for PBRM1 was 0.1. Please clarify for the non-statisticians among us the statistical significance of the association and how the 95% CIs do not overlap with 1 if the p-values are > 0.05.
8. The authors report that "After adjusting for VHL status, PBRM1 mutational status did not show an independent relationship with OS (HR = 0.68, 95% CI: 0.44 - 1.03)." whereas values without adjustment for VHL status (HR = 0.64, 95% CI: 0.42 - 0.97) are nearly identical. This might be more accurately stated "After adjusting for VHL status, the association of PBRM1 mutational status with OS no longer met statistical significance" Similarly, in the Discussion, "After adjusting for VHL status we did not find support for an independent association between other driver mutations and survivorship." should be "we did not detect a statistically significant independent association..."
9. In Figure 8 there was a clear separation of curves by SV count with log-rank P=0.004, so it is unclear how after adjusting for VHL status, the HR for the association of higher SV count with OS was only 1.01 (95% CI: 1.00 - 1.10). Can the authors please clarify the statistics here?
10. The term "drug repurposing" (as used in the Abstract and Discussion) most commonly refers to

applying non-cancer drugs to a cancer indication, not applying drugs targeting a specific biomarker in one cancer type to another cancer type. Usually this would be called expanding the indication, so in the Discussion "Examples of drugs that might be repurposed for treating ccRCC include:" would be more accurately "Examples of drugs where the population of interest could be expanded to include ccRCC patients include:"

11. Similarly, "survivorship" generally refers to the patient state after completion of cancer treatment, so "increased survivorship" should be "prolongation of survival."

12. In the discussion, the authors claim to "validate pTERT mutations as drivers, thereby further substantiating telomerase dysfunction in the development of ccRCC." However, there are multiple prior studies demonstrating the role of pTERT mutations in ccRCC, so it is unclear what additional "validation" was performed here as opposed to "re-demonstration".

Reviewer #3:

Remarks to the Author:

In this paper, Culliford et al. reported an interesting data set of whole genome sequencing on 778 clear cell renal cell carcinoma in Gel cohort. Through integrated analysis, they identified new driver genes, especially those in epigenomic regulation, and potentially targetable therapies. They further correlated genomic features with clinical outcomes and found that increased SV was associated with worse survival, but VHL mutation and four specific copy number gains were correlated with better OS. This manuscript contains an abundance of genomic profiling data from ccRCC samples and represents an important resource for the field. The figures are generally well prepared. However, I have several concerns about this current version of the paper.

Major concerns:

1. The main limitation of the study is that despite the large-scale WGS analysis of ccRCC tumors available, the detailed clinicopathological features (eg. age, sex, symptoms at diagnosis, smoking history, family history, ECOG performance status, BMI, TNM, sarcomatoid component, surgery, systematic therapeutic regimen et.al.) of individual were lacking, thus hinder further analysis of the correlation between genomic features and clinicopathological characteristics.
2. The author use OS for survival analysis. However, OS is influenced by many factors related not only to tumor. In sub-analysis, patients > 70 years exhibited worst OS. That may be associated with increased comorbidities in this setting, but not RCC-specific death. Using cancer-specific survival for these analysis would be suitable.
3. Page 8, line 236-239, 37 patients received IO-based therapy in the cohort. How about the treatment regimens (IO monotherapy, IO+IO, IO+TKIS)? Different immunotherapeutic approaches would interfere analysis. "21 of these there was documented evidence of clinical benefit". What are the evaluation criteria and primary outcomes for clinical benefit?
4. The authors report a APOBEC signature in some tumors. However, it is unclear that mutations were found in AID/APOBEC genes. Also, the authors found three patients with dMMR. Did they further validate the dMMR status of these tumors, since dMMR was less common in RCC. Did any of these patients with dMMR develop metastatic disease and were they treated with checkpoint inhibitors?
5. Page 3, line 84-86, "twenty-two of the patients (2.8%) were carriers 84 of pathogenic germline variants". How about family history and extrarenal manifestations of these patients?
6. A subsets of patients < 45 years were included in this study. How about the genomic features of ccRCC in this younger population?

7. Sub-analysis regarding the mutational signature with clinical features and survival outcome needed to be added.

8. Previous study demonstrated RCC had the highest proportion of indel mutations which would induce tumor-specific neoantigens with high affinity (PMID: 28694034). Did the authors find the relationship between indel burden and tumor-specific neoantigen burden, HLA status and CD8 infiltrations?

9. What is ccRCC diagnostic criteria used in this study? ELOC-mutated RCC was defined as distinct RCC entity according to the 2022 World Health Organization Classification. Also, a subsets of RCC with TCS genes mutation were now categorized to eosinophilic solid and cystic (ESC) RCC.

Minor concerns:

1. The author needed to add a table about the detail baseline characteristics of individual.

2. Page 5, line 125, "93 unique alterations were targetable (OncoKB Level 1-4)". But in Table S7, there were 65 targetable alterations. Also, targetable alterations in Table S7 were incompatible with those in Figure S3B.

3. Page 8, line 217-219, Figure 8 fails to reflect this result. Also, these results could not be found in Fig S6, Table S15 and Table S16.

4. Figure 3 legends (B), MAPK should be changed to PI3K/AKT/MTOR.

5. Figure 6 A and B, BAP1 mutation needed be added in these figures.

6. Figure S7D, two patients with ISUP grade 1 recurred early. Did they have special mutations?

Reviewer #4:

Remarks to the Author:

This is a very important study that covers whole genome sequencing of 778 ccRCC primary cancer specimens. The analysis is comprehensive. There are some minor things that need to be revised:

General:

1) Figure citations in the text: please cite each sub-panel in the text where the figures are referenced such as Fig. 3a, Fig. 3b etc.

2) Carefully go through the grammar errors: comma (,) is missing in many places (line 54; line 57; line 131; line 175; line 194; line 198; line 202.....;

several places have grammar issues:

a. Abstract: line 33-34, the twin observations support the rationale for immunotherapy.

b. Intro: line 50, Mixed outcomes of demonstrate that;

c. Line 96: sentence is fragmented and wrongly placed comma;

d. Line 1223-126: rewrite the sentence;

e. Line: 167: none of which was

Specific related to science:

3) Please state the result based on the data for accuracy. In the abstract: the only significant correlation is between SV and OS based on Fig. 8. VHL, TCRA, and others are not significantly correlated with OS. Rewriting based on the factual data should be done in abstract and the text referencing Fig 8.

4) Please replace Immune checkpoint inhibitors to common abbreviation (ICI).

5) Line 72: define PCR-free and why it is important here.

6) Line 82: Median tumour mutational burden was 1.88/Mb, based on Supple Table 2. That is for none-coding mutations. Should the coding mutation median rate of 2.32 cited here?

7) Line 84-96: please list genes in the RCC susceptibility or other cancer susceptibility gene. Please change another to other.

8) There are several places in the paper that may be wrongly stated in the correlation of OS and other parameters:

a. Line 218-219: Fig 8a, $P = 0.2$, indicating no correlation between VHL mutations and OS;

b. Line 224-225: Higher SV count was associated with worse OS, but the HR ratio is labeled as 1.01, 95% CI: 1-1:10). How can that be significant with close to 1 HR ratio?

c. TCRA T cell data, Fig. 8d. $P = 0.2$. Why is this stated as associated with a better OS?

Reviewer comments

Reviewer #1, expertise in ccRCC omics (Remarks to the Author):

Culliford et al. analyzed whole genome sequencing (WGS) data from 778 ccRCC patients from the UK Genomics England 100,000 Genomes Project. This study provides a comprehensive picture of the genomic changes in ccRCC tumors. Especially, this study revealed some novel cancer drivers for ccRCC, even though the mutation frequencies for these potential drivers are low. Additionally, they have found associations between various genomic characterizations of the ccRCC patients/tumors and the clinical outcomes. This is a nicely performed, large scale, and comprehensive genomic analyses of ccRCC that will serve as a valuable resource for the research community. However, the claimed novel drivers were not carefully examined informatically and have not been experimentally validated. Furthermore, clinical associations shown in the manuscript are marginally significant and need careful evaluation and validation. These two aspects need to be addressed by the authors to demonstrate the value of this study.

There are a few additional specific comments and concerns.

1. Although the value of these WGS data and analyses as references is clear, most of the findings are confirming similar ones from various prior WES studies of ccRCC.

Response: Inevitably with any new cancer sequencing project the probability of identifying a novel common driver is unlikely. However, our study has provided the opportunity to identify novel cancer driver genes mutated at low frequency, which in other studies such as TCGA have not been powered to implicate the genes as drivers (although in most cases the mutational frequencies are consistent; **Supplementary Figure 2**). Importantly the novel drivers we identify emphasise the central importance of epigenetic modification in the development of ccRCC through SWI/SNF mediated chromatin remodelling (*BRD3*, *ARID2*), histone deubiquitination (*CUL3*, *FBXW7*) as well as the role of methylcytosine dioxygenase activity (*TET1*).

2. The novel drivers are potentially interesting in illustrating the etiology and key molecular events driving the progression of ccRCC. However, it may not be easy to determine if these are truly important drivers for ccRCC due to the low frequency and the absence of any functional studies.

Response: We acknowledge this point. While functional studies are outside the scope of the current analysis we state Mutations in the novel drivers were frequently accompanied by loss of heterozygosity (LOH) (see "Recurrent structural and copy number alterations"), implying loss of function (**Supplementary Fig. 3**). Furthermore, leveraging TCGA (Weinstein et al. 2013), GTEx (Lonsdale et al. 2013) and DepMap (Tsherniak et al. 2017; Behan et al. 2019) data as well as referencing literature provide supporting evidence that 6/13 of the novel genes we identify have relevance to the biology of RCC.

3. ccRCC drivers have long been established and the order of the events have also been described in several major studies. How do the claimed new drivers fit into the big picture of ccRCC oncogenesis?

Response: To complement **Fig. 2** we now provide further text to contextualise findings.

4. To gain a complete understanding of ccRCC drivers, the authors can combine TCGA, CPTAC, and their own WGS-based ccRCC data for a comprehensive driver discovery.

Response: To contextualise our findings we now include information on the driver analysis of TCGA based on exome sequencing data and the mutation frequency of selected genes reported by panel testing of TRACER-X (Revised text and **Supplementary Fig. 3**).

5. I would also like to see more detailed analysis of structural events using WGS data. Have a clear map of structural events and the heterogeneity of such events in tumors would be important for realizing the value of the WGS data.

Response: As requested we now provide structural variant breakpoint plots, segregated by SV type, for SV hotspot regions in the Supplementary information (**Supplementary Fig. 6-8**).

6. Have the authors discovered any non-coding events from WGS potentially associated with ccRCC pathogenesis? This will provide novelty to this study.

Response: In our original submission we used OncodriveFML, ActiveDriverWGS and single-base-substitution negative binomial regression to identify non-coding drivers, requiring the regions to be declared significant if significant by all three algorithms. In retrospect we appreciate that this approach is too conservative and we reanalysed our data adopting a different strategy. To combine *P*-values from the three methods, which are not necessarily independent, we adopted a similar strategy as (Gerstung et al. 2020) using Empirical Brown's method (Poole et al. 2016). We adjusted for multiple-testing using the Benjamini-Hochberg procedure (Benjamini and Hochberg 1995). Post filtering of significant regions (i.e. Brown's *Q*-value < 0.001) was conducted to exclude those with accumulation of mutations caused by sequencing artefacts or mutational processes, by implementing the following inclusion criteria: (i) >2 mutations being present within the region; (ii) >50% of mutations located in a mappable genomic region (CRG alignability, DAC blacklisted, regions, and DUKE uniqueness) and (v) <50% of mutations resulting from APOBEC or AID signatures. Empirical Brown's method is predicated on the assumption that *P*-values from each method approximately follow a scaled χ^2 -distribution, which is however not necessarily the case with the output from the three algorithms. Acknowledging this, for reasons of pragmatism, we conservatively only discuss regions found to be significantly mutated by at least two of the three methods.

7. Have the authors considered generating some long reads WGS data to perform a better map of some of the breakpoints? This will provide novelty to this study.

Response: We acknowledge that generating long read sequencing data would complement the current short read sequencing, especially with respect to deconvoluting breakpoints. Unfortunately, however this is not the agenda presently for Genomics England.

Minor points:

1. "Twenty-two of the patients (2.8%) were carriers of pathogenic germline variants in one of the well-established RCC susceptibility genes and 10 (1.2%) were carriers of a variant in another cancer

susceptibility gene (Supplementary Table 3).” The description of the genes involved should be more specific.

Response: As requested we now provide a specific description of genes.

2. Line 192 – “TMB” should be defined at its first use (Tumor mutational Burden).

Response: Error corrected.

3. The claim of “only 9% of tumors were predicted to exhibit some form of genetically-driven immune evasion” needs stronger, clear evidence.

Response: To clarify matters we now state this refers to HLA LOH, APG inactivation or HLA somatic mutation.

4. The ordering of mutational events section lacks details. Please expand.

Response: As requested we have expanded this section in both the main manuscript and the supplementary methods. Specifically, in the main text we now additionally state “These methods estimate the time at which copy number gains occur by considering the fraction of mutations before and after the gain. In the regions of copy number gain individual mutations can therefore be estimated to be early or late events; subclonal mutations will, by definition, have occurred late in tumorigenesis.”

Reviewer #2, expertise in ccRCC omics (Remarks to the Author):

Culliford, et al. report results from whole genome sequencing of 778 ccRCC patients enrolled in the 100,000 Genomes Project in relation to Driver mutations, Recurrent structural and copy number alterations, Mutational signatures, Ordering of mutational events, Immune evasion and Clinico-pathological relationships. Reassuringly, driver mutations are consistent with what has been previously reported in ccRCC, and the authors report 13 events which either had not been reported previously or have frequencies <1% in landmark genomics studies of ccRCC. The authors confirm the importance of TERT promoter mutations in RCC, demonstrate the smoking mutational signature in a subset of cases, and demonstrate that only a minority of cases harbor alterations that would predict for lack of response to checkpoint immunotherapy. Overall, this is a large and comprehensive data set that is of interest to the renal cell carcinoma community, though the findings largely confirm existing understanding of ccRCC biology rather than provide novel insights.

I have the following largely minor comments:

1. In Figure 2, the authors highlight in red mutations that had previously been reported in <1% of patients in other RCC cohorts, but are there gene mutations detected in this study that have not been previously reported in RCC at all?

Response: Fig. 2 does not refer to individual “mutations”, but the number of carriers of a nonsynonymous mutation in a driver gene. To avoid ambiguity we have revised the legend text.

2. In Figure 3, does "non-ccRCC" mean RCC of other histologies, or does it mean in cancers other than RCC? The abbreviation could be changed, or this could be clarified in the legend.

Response: We have revised the figure legend to avoid ambiguity. We now state “non-ccRCC” drivers refers to drivers for any cancer other than RCC.

3. Figure 5 would benefit from a figure legend. I am assuming that the numbers on the X-axis refer to the number of samples that have evidence of each signature, and that the Y-axis refers to somatic mutations per Mb in those samples, but it would help the reader to 1) understand what these signatures are (possibly with a Table below the figure) and 2) what it represents that certain signatures are associated with higher or lower somatic mutations/Mb.

Response: We have revised the legend for Fig. 5.

4. The authors report "three tumours displayed a hypermutated phenotype: i.e., excessively high SNV/indel mutation burden 83 (maximal SNV/Mb = 33.65, maximal indel/Mb = 21.77)." and also note that "Three cases displayed evidence of dMMR, of which two harboured MLH1 somatic mutations accompanied by LOH, but none carried germline pathogenic MMR variants. Two of the cases had mutations assigned to signatures associated with dMMR (SBS20, SBS26)."

Was there overlap between the hypermutated cases and those with dMMR? If so, this should be stated, and if not, why the discrepancy?

Response: We have edited the text when describing mutational signatures to explicitly state that both cases with dMMR signatures had a hypermutated phenotype.

5. In Figure 6c, please clarify what the x-axis of Log(OR) represents. There seems to be a discrepancy in that TP53 mutations are more likely to be subclonal (with OR < 1 in Figure 6B) but are early events (Figure 6C) - can the authors comment on this?

Response: We have completely revised Fig. 6 in response to the reviewers comment. Fig. 6a now shows the substructure of clonal mutations (early, late, unknown) which are used when determining the order of driver events. Fig. 6b shows the odds ratios for subclonality for each driver gene relative to the distribution for all driver genes. We also now report the mean ordering in Fig. 6c rather than the Log(OR) to facilitate interpretation. Specifically regarding TP53, we note that while mutations are more likely to be subclonal, although not statistically significant, the large proportion (33%) and raw number (11) of early clonal mutations results in TP53 appearing early in the league model.

6. In regards to the immune landscape, it would be interesting to know if ccRCC with sarcomatoid features has a different landscape than those without since sarcomatoid RCCs have more clinical benefit from immune checkpoint inhibitors. Is the presence of sarcomatoid features known in these cases?

Response: As requested we now provide an subanalysis for sarcomatoid features. We now state that while sarcomatoid features were not associated with a difference in genetically predicted immune escape. However, although not statistically significant but consistent with other work, sarcomatoid histology was associated with improved response to ICI (**Supplementary Table 21**).

7. In Figure 8, the Log-rank P value for OS for VHL mutation was 0.2 and for PBRM1 was 0.1. Please clarify for the non-statisticians among us the statistical significance of the association and how the 95% CIs do not overlap with 1 if the p-values are > 0.05.

Response: We apologise for any ambiguity here. We now only include log rank association test results in the figures. All associations between survival and genomic features reported in the text are based on Cox regression, adjusting for covariates.

8. The authors report that "After adjusting for VHL status, PBRM1 mutational status did not show an independent relationship with OS (HR = 0.68, 95% CI: 0.44 - 1.03)." whereas values without adjustment for VHL status (HR = 0.64, 95% CI: 0.42 – 0.97) are nearly identical. This might be more accurately stated "After adjusting for VHL status, the association of PBRM1 mutational status with OS no longer met statistical significance" Similarly, in the Discussion, "After adjusting for VHL status we did not find support for an independent association between other driver mutations and survivorship." should be "we did not detect a statistically significant independent association..."

Response: We acknowledge this point and have revised our text accordingly.

9. In Figure 8 there was a clear separation of curves by SV count with log-rank P=0.004, so it is unclear how after adjusting for VHL status, the HR for the association of higher SV count with OS was only 1.01 (95% CI: 1.00 - 1.10). Can the authors please clarify the statistics here?

Response: See Response to point 7.

10. The term "drug repurposing" (as used in the Abstract and Discussion) most commonly refers to applying non-cancer drugs to a cancer indication, not applying drugs targeting a specific biomarker in one cancer type to another cancer type. Usually this would be called expanding the indication, so in the Discussion "Examples of drugs that might be repurposed for treating ccRCC include:" would be more accurately "Examples of drugs where the population of interest could be expanded to include ccRCC patients include:"

Response: We have revised our text as per recommendation to now state "Examples of drugs where the population of interest could be expanded to treat ccRCC include:".

11. Similarly, "survivorship" generally refers to the patient state after completion of cancer treatment, so "increased survivorship" should be "prolongation of survival."

Response: As requested we have revised our text replacing "survivorship" with "patient overall survival" and "increased survival".

12. In the discussion, the authors claim to "validate pTERT mutations as drivers, thereby further substantiating telomerase dysfunction in the development of ccRCC." However, there are multiple prior studies demonstrating the role of pTERT mutations in ccRCC, so it is unclear what additional "validation" was performed here as opposed to "re-demonstration".

Response: We have revised the text.

Reviewer #3, expertise in ccRCC genomics and mutational signatures (Remarks to the Author):

In this paper, Culliford et al. reported an interesting data set of whole genome sequencing on 778 clear cell renal cell carcinoma in Gel cohort. Through integrated analysis, they identified new driver genes, especially those in epigenomic regulation, and potentially targetable therapies. They further correlated genomic features with clinical outcomes and found that increased SV was associated with worse survival, but VHL mutation and four specific copy number gains were correlated with better OS. This manuscript contains an abundance of genomic profiling data from ccRCC samples and represents an important resource for the field. The figures are generally well prepared. However, I have several concerns about this current version of the paper.

Major concerns:

1. The main limitation of the study is that despite the large-scale WGS analysis of ccRCC tumors available, the detailed clinicopathological features (eg. age, sex, symptoms at diagnosis, smoking history, family history, ECOG performance status, BMI, TNM, sarcomatoid component, surgery, systematic therapeutic regimen et.al.) of individual were lacking, thus hinder further analysis of the correlation between genomic features and clinicopathological characteristics.

Response: Some of the variables mentioned such as age, sex and ICI-type systemic therapeutic regimens were already incorporated into the first draft of the manuscript. We have since added the presence of sarcomatoid as a variable, and a multi-regression sub-analysis on TCRA T-cell fraction. No information on smoking history, family history, ECOG performance status and BMI had been collected by Genomics England allowing us to examine the relationship between genomic features and these phenotypes.

2. The author use OS for survival analysis. However, OS is influenced by many factors related not only to tumor. In sub-analysis, patients > 70 years exhibited worst OS. That may be associated with increased comorbidities in this setting, but not RCC-specific death. Using cancer-specific survival for these analyses would be suitable.

Response: Since we submitted our paper additional data on cancer specific survival (CSS) has become available through Genomics England and we now provide an analysis of the relationship between genomic features and CSS.

3. Page 8, line 236-239, 37 patients received IO-based therapy in the cohort. How about the treatment regimens (IO monotherapy, IO+IO, IO+TKIS)? Different immunotherapeutic approaches

would interfere analysis. “21 of these there was documented evidence of clinical benefit”. What are the evaluation criteria and primary outcomes for clinical benefit?

Response: We now state in the methods that progression free-survival was defined as survival time without disease progression on current systemic treatment for ≥ 6 months and that disease progression was defined as per RECIST 1.1 (*i.e.* at least a 20% increase in the sum of diameters of (target) lesions).

4. The authors report a APOBEC signature in some tumors. However, it is unclear that mutations were found in AID/APOBEC genes. Also, the authors found three patients with dMMR. Did they further validate the dMMR status of these tumors, since dMMR was less common in RCC. Did any of these patients with dMMR develop metastatic disease and were they treated with checkpoint inhibitors?

Response: We have revised the text reporting the number of patients with an APOBEC signature and a mutation in an APOBEC gene. We also state that two of the cases with dMMR have a hypermutated phenotype, and that they have somatic mutations in dMMR genes. We report that none of these cases were treated with checkpoint inhibitors and did not develop metastatic disease.

5. Page 3, line 84-86, “twenty-two of the patients (2.8%) were carriers of pathogenic germline variants”. How about family history and extrarenal manifestations of these patients?

Response: We now state that four of the patients who were carriers of a cancer susceptibility gene had a previous history of a non-RCC cancer (prostate, CLL, testicular and thyroid cancers). Unfortunately family history of cancer was not routinely collected on patients as part of the Genomics Program.

6. A subsets of patients < 45 years were included in this study. How about the genomic features of ccRCC in this younger population?

Response: We now provide an analysis and commentary on the genomic landscape of ccRCC in younger patients (**Supplementary Table 12**).

7. Sub-analysis regarding the mutational signature with clinical features and survival outcome needed to be added.

Response: We now provide an analysis of the relationship between mutational signatures and survival outcome.

8. Previous study demonstrated RCC had the highest proportion of indel mutations which would induce tumor-specific neoantigens with high affinity (PMID: 28694034). Did the authors find the relationship between indel burden and tumor-specific neoantigen burden, HLA status and CD8 infiltrations?

Response: In **Supplementary Table 14**, indel burden was found to be associated with neoantigen-burden. We have done additional analysis involving HLA status (**Supplementary Table 16**) and used TCRA T-cell infiltration (**Supplementary Table 13**) as a proxy for CD8+ infiltration. We found no association between mutational burden (total, SNV or indel) with HLA status and CD8+ infiltration.

9. What is ccRCC diagnostic criteria used in this study? ELOC-mutated RCC was defined as distinct RCC entity according to the 2022 World Health Organization Classification. Also, a subsets of RCC with TCS genes mutation were now categorized to eosinophilic solid and cystic (ESC) RCC.

Response: The histology of the ccRCC cohort was reported before the World Health Organisation (WHO) Classification 5th edition (2022) was released, with the WHO Classification of Urinary and Male Genital tumours 3rd or 4th edition applied to the cohort (the two versions having identical ccRCC diagnostic criteria). Manual inspection of available histology reports typically have a lack of proportional based information regarding tumour cellular structures, with mentions of identifying an eosinophilic cell structure being used primarily to identify the core RCC subtypes, for example differentiating chromophobe RCC from renal oncocytoma. Nevertheless, we still report the existence of *ELOC*-mutated tumours, *ELOC* being a driver gene, and how they are mutually exclusive, with zero co-occurring mutations, to *VHL*-mutated tumours.

Minor concerns:

1. The author needed to add a table about the detail baseline characteristics of individual.

Response: Summary level data on all the patients is provided in **Supplementary Table 1**. Unfortunately we cannot provide per-individual characteristics as this will breach Genomic England's confidentiality policy (i.e information that could be used to identify a participant in the programme).

2. Page 5, line 125, "93 unique alterations were targetable (OncoKB Level 1-4)". But in Table S7, there were 65 targetable alterations. Also, targetable alterations in Table S7 were incompatible with those in Figure S3B.

Response: Thank you for noticing this error. The correct number is 60 unique OncoKB annotated alterations. For clarification, we now provide two extra columns in Table S7, stating the number of total mutations found for each gene/mutation. **Supplementary Figures 9a and 9b** (formerly 3a and 3b) refers to the number of patients with at least one mutation in one of the targetable genes in the figure, and not the full number of targetable alterations itself.

3. Page 8, line 217-219, Figure 8 fails to reflect this result. Also, these results could not be found in Fig S6, Table S15 and Table S16.

Response: All information requested now provided.

4. Figure 3 legends (B), MAPK should be changed to PI3K/AKT/MTOR.

Response: Text changed to reflect this.

5. Figure 6 A and B, BAP1 mutation needed be added in these figures.

Response: BAP1 now added to these figures.

6. Figure S7D, two patients with ISUP grade 1 recured early. Did they have special mutations?

Response: The two patients did not have any specific features.

Reviewer #4, expertise in immunogenomics for ccRCC (Remarks to the Author):

This is a very important study that covers whole genome sequencing of 778 ccRCC primary cancer specimens. The analysis is comprehensive. There are some minor things that need to be revised:

General:

1) Figure citations in the text: please cite each sub-panel in the text where the figures are referenced such as Fig. 3a, Fig. 3b etc.

Response: We now cite each sub-panel for all figures and supplementary figures, and made minor alterations to where each subpanel is cited.

2) Carefully go through the grammar errors: comma (,) is missing in many places (line 54; line 57; line 131; line 175; line 194; line 198; line 202.....);

several places have grammar issues:

a. Abstract: line 33-34, the twin observations support the rationale for immunotherapy.

b. Intro: line 50, Mixed outcomes of demonstrate that;

c. Line 96: sentence is fragmented and wrongly placed comma;

d. Line 1223-126: rewrite the sentence;

e. Line: 167: none of which was

Response: Grammatical errors corrected.

Specific related to science:

3) Please state the result based on the data for accuracy. In the abstract: the only significant correlation is between SV and OS based on Fig. 8. VHL, TCRA, and others are not significantly correlated with OS. Rewriting based on the factual data should be done in abstract and the text referencing Fig 8.

Response: As per Reviewer 2 point.

4) Please replace Immune checkpoint inhibitors to common abbreviation (ICI).

Response: The common abbreviation ICI is now used throughout the text.

5) Line 72: define PCR-free and why it is important here.

Response: We now state the definition in the manuscript and supplementary methods.

6) Line 82: Median tumour mutational burden was 1.88/Mb, based on Supple Table 2. That is for none-coding mutations. Should the coding mutation median rate of 2.32 cited here?

Response: We have now state the median tumour mutational burden as 2.07/Mb

7) Line 84-96: please list genes in the RCC susceptibility or other cancer susceptibility gene. Please change another to other.

Response: Genes are now explicitly stated in the text. As requested the word "another" has been changed.

8) There are several places in the paper that may be wrongly stated in the correlation of OS and other parameters:

- a. Line 218-219: Fig 8a, $P = 0.2$, indicating no correlation between VHL mutations and OS;
- b. Line 224-225: Higher SV count was associated with worse OS, but the HR ratio is labeled as 1.01, 95% CI: 1-1:10). How can that be significant with close to 1 HR ratio?
- c. TCRA T cell data, Fig. 8d. $P = 0.2$. Why is this stated as associated with a better OS?

Response: a. and c.: As stated in the third comment made by this reviewer, this has been rectified.
b.: We have changed "95% CI: 1 - 1.10" to "95% CI: 1.00 - 1.10" to make it clearer that the hazard ratio confidence interval does not overlap 1.

Reviewers' Comments:

Reviewer #1:

Remarks to the Author:

The authors have addressed the concerns I had. The manuscript is much improved.

Reviewer #2:

Remarks to the Author:

The authors performed several additional analyses to address reviewers comments and made the recommended edits so I have no additional comments.

Reviewer #3:

Remarks to the Author:

My queries have been largely addressed and I support moving towards publication.

Reviewer #4:

Remarks to the Author:

My comments are fully addressed. No further concern.